# Chronic Liver Disease Increases Mortality Following Pancreatoduodenectomy

**DOI:** 10.3390/jcm10112521

**Published:** 2021-06-07

**Authors:** Jana Enderes, Jessica Teschke, Steffen Manekeller, Tim O. Vilz, Jörg C. Kalff, Tim R. Glowka

**Affiliations:** Department of Surgery, University Hospital Bonn, 53127 Bonn, Germany; jana.enderes@ukbonn.de (J.E.); jessica.teschke@ukbonn.de (J.T.); steffen.manekeller@ukbonn.de (S.M.); tim.vilz@ukbonn.de (T.O.V.); kalff@uni-bonn.de (J.C.K.)

**Keywords:** pancreaticoduodenectomy, Whipple, delayed gastric emptying, cirrhosis, fibrosis

## Abstract

According to the International Study Group of Pancreatic Surgery (ISGPS), data about the impact of pre-existing liver pathologies on delayed gastric emptying (DGE) after pancreatoduodenectomy (PD) according to the definitions of the International Study Group of Pancreatic Surgery (ISGPS) are lacking. We therefore investigated the impact of DGE after PD according to ISGPS in patients with liver cirrhosis (LC) and advanced liver fibrosis (LF). Patients were analyzed with respect to pre-existing liver pathologies (LC and advanced LF, *n* = 15, 6% vs. no liver pathologies, *n* = 240, 94%) in relation to demographic factors, comorbidities, intraoperative characteristics, mortality and postoperative complications, with special emphasis on DGE. DGE was equally distributed (DGE grade A, *p* = 1.000; B, *p* = 0.396; C, *p* = 0.607). Particularly, the first day of solid food intake (*p* = 0.901), the duration of intraoperative administered nasogastric tube (NGT) (*p* = 0.812), the rate of re-insertion of NGT (*p* = 0.072), and the need for parenteral nutrition (*p* = 0.643) did not differ. However, patients with LC and advanced LF showed a higher ASA (American Society of Anesthesiologists) score (*p* = 0.016), intraoperatively received more erythrocyte transfusions (*p* = 0.029), stayed longer in the intensive care unit (*p* = 0.010) and showed more intraabdominal abscess formation (*p* = 0.006). Moreover, we did observe a higher mortality rate amongst patients with pre-existing liver diseases (*p* = 0.021), and reoperation was a risk factor for higher mortality (*p* ≤ 0.001) in the multivariate analysis. In our study, we could not detect a difference with respect to DGE classified by ISGPS; however, we did observe a higher mortality rate amongst these patients and thus, they should be critically evaluated for PD.

## 1. Introduction

Liver cirrhosis (LC) is the result of chronic liver injury by various stimuli such as chronic hepatitis B and C virus infections, alcohol-related liver disease and non-alcoholic fatty liver disease (NAFLD) [1,2,3,4,5]. Chronic liver injury leads to structural changes involving the activation of hepatic stellate cells and the production of the extracellular matrix, which accumulates and leads to fibrotic scarring, and subsequently to LC if left untreated [6]. End-stage LC and its complications—such as variceal bleeding, ascites, spontaneous bacterial peritonitis, or hepatic encephalopathy—can be life-threatening [7]. The prevalence of LC has evolved in recent years due to an increasing number of people showing the above-mentioned risk factors that can lead to LC [8]. This automatically results in an increased number of unrecognized (and only intraoperatively diagnosed) advanced LF and LC in patients requiring PD [9]. Patients with LC show high mortality and morbidity after surgery [10]. The increased risk for high mortality is due to increased bacterial infections and compromised wound healing, poor nutritional status, increased intra- and postoperative bleeding, and hydropic decompensation due to ascites, peripheral edema and postoperative renal failure [11,12,13]. The postoperative outcome also depends on the specific procedure, with mortality rates ranging from 0.6 to 1.3% for laparoscopic cholecystectomy, and from 14 to 29% for colorectal surgery [14]. As for PD, postoperative mortality ranges from 0 to 55% depending on the Child–Pugh status [15,16,17,18,19].

The data regarding gastric motility and gastroparesis in patients with LC are inconclusive. The majority of studies report impaired gastric motility in patients with LC per se [20,21,22,23]. There are two studies showing no difference in gastric emptying time [24,25]. However, since both of these studies only used liquid test meals, these data should be interpreted cautiously. Due to the evolving number of cirrhotic patients undergoing PD, the question arises whether these patients are put at an additional risk in respect to DGE after PD. The aim of this study was to investigate the impact of common specific complications after PD according to the ISGPS definitions, with special emphasis on DGE in patients with LC and advanced LF after PD.

## 2. Materials and Methods

We retrospectively studied all patients that underwent PD at our department between January 2012 and December 2020 (*n* = 275). Patients who had previously had gastric resection (*n* = 2) and patients with prolonged postoperative resumption to normal diet—independently of DGE (long-term ventilation >7 days, fasting due to PF) (*n* = 18)—were excluded. The remaining total of 255 patients were included in our study. All pancreatic resections were prospectively recorded in a pancreatic resection database with the approval of the institutional ethics committee (ethics committee of the Rheinische Friedrich-Wilhelms University Bonn, 347/13) and after obtaining written informed consent from the participants. Patients were analyzed with respect to pre-existing liver pathologies (LC and advanced LF, *n* = 15, 6% vs. no liver pathologies, *n* = 240, 94%) in relation to demographic factors, comorbidities, intraoperative characteristics, hospital stay, morbidity and mortality, and postoperative complications, with special emphasis on DGE. Morbidity and mortality were documented according to the Clavien–Dindo classification [26]. PF, PPH and DGE were classified according to the definitions of the International Study Group on Pancreatic Surgery [27,28,29]. LC and advanced LF were diagnosed by ultrasound elastography, high-resolution imaging (CT scan or MRI; typical signs include surface nodularity, enlargement of liver segment 1 and signs of portal hypertension) and/or intraoperatively by macroscopy and/or histology.

Perioperative management was carried out according to our institution’s standard operating procedure protocol. In cases of a potential high-malignant tumor, patients were discussed by our multidisciplinary Tumor Board. If considered for surgery, bowel preparations were not administered and patients were allowed to eat solid food up to 6 h before surgery, and to drink liquids up to 2 h before surgery. In case of malnutrition, sip feeds were administered at least one week prior to surgery.

PD was performed by 4 certified senior pancreatic surgeons (JCK, SM, TRG, NS). Resection was carried out in a standardized fashion and for reconstruction, a single loop technique with ante- or retrocolic reconstruction was used. In the case of retrocolic reconstruction, supra- or infracolic routes were used [30]. Duodenoenterostomy, pancreatogastrostomy and end-to-end choledochojejunostomy were carried out as previously described [31,32]. A classic Whipple procedure with double-loop reconstruction was only carried out if the antrum was directly infiltrated. Perioperatively, a mid-thoracic peridural catheter and a 14 French nasogastric tube (NGT) were placed by default. In the case of contraindications for a peridural catheter, opioids were given via a patient-controlled analgesia. Before closure of the abdomen, two soft drains were placed at the sites of pancreatogastrostomy and choledochojejunostomy.

Postoperatively, patients stayed at the intensive care unit for at least one day and were allowed to directly drink water. Patients then transitioned to a normal diet followed by easily digestible/fat-reduced meals on POD3, easily digestible/fiber-reduced meals on POD4, a basic diet (no pulses/no brassica) on POD5, and a normal diet on POD6. NGT was removed if daily secretions were less than 500 mL, and soft drains were subsequently removed if amylase levels in the drainage fluid were normal on POD3. In the case of pancreatic fistula, octreotide (100 μg 3×/d s.c.) was administered for 5 days. If patients had not shown bowel movement by POD2, an oral laxative (magnesium sulfate) was administered. In the case of vomiting, transition to a normal diet was discontinued and an NGT was re-inserted. All patients received an antibiotic prophylaxis with an aminopenicillin plus β-lactamase inhibitor and a weight-adapted thrombosis prophylaxis.

Data were recorded and analyzed with Excel 2013 (Microsoft Corporation, Redmond, Washington, DC, USA) and SPSS 24 (IBM Corporation, Armonk, New York, NY, USA). Statistical analyses were carried out as described in one of our previous studies [32]: continuously and normally distributed variables were expressed as medians ± standard deviation and analyzed using the Student’s *t* test, while non-normally distributed data were expressed as medians and interquartile range, and analyzed using the Mann–Whitney U test. Categorical data were expressed as proportions and compared with either the Pearson x^2^ test or the Fisher’s exact test, as appropriate. Factors with *p* < 0.1 in the univariate analysis were included in multivariate stepwise logistic regression analysis, with a significance level of *p* < 0.05 for inclusion and *p* < 0.10 for removal in each step. The relative risk was described by the estimated odds ratio, with 95% confidence intervals. A *p*-value < 0.05 was considered statistically significant.

## 3. Results

Of the 255 patients undergoing pancreatoduodenectomy, 15 showed pre-existing liver cirrhosis (LC) or advanced liver fibrosis (LF) (LC, *n* = 5; LF, *n* = 10). Of the former, three were classified as CHILD A and two were classified as CHILD B. CHILD A cirrhosis was caused by alcohol abuse (*n* = 1), chronic hepatitis C (*n* = 1) and non-alcoholic fatty liver disease (NASH) (*n* = 1), whereas CHILD B cirrhosis was caused by primary biliary cirrhosis (PBC) (*n* = 1) and NASH (*n* = 1). Advanced fibrosis was caused by polycystic liver disease (*n* = 3), alcohol abuse (*n* = 2), chronic hepatitis B and C (*n* = 1), chronic hepatitis C on its own (*n* = 1), hepatic sinusoidal obstruction syndrome (*n* = 1), NASH (*n* = 1) and drug-related inducement (*n* = 1).

Patients with LC and advanced LF were comparable to patients without pre-existing liver pathologies in regard to demographics and preoperative data (Table 1); however, patients with LC and advanced LF showed a higher frequency of alcohol abuse (53% vs. 26%, *p* = 0.036), and significantly more comorbidities measured by a higher Charlson Morbidity Index (CCI) (CCI 4 (3–5) vs. CCI 2 (2,3), *p* ≤ 0.001) were observed among them (Table 1). Moreover, they were classified as being at higher perioperative risk, as represented by a higher ASA (American Society of Anesthesiologists) score (ASA 3 (2,3) vs. ASA 2 (2,3), *p* = 0.016).

Intraoperative data such as surgery duration and technical aspects did not differ between the two groups (Table 1). However, we did observe more erythrocyte concentrates being transfused in patients with LC and advanced LF (2 (0–3) vs. 0 (0–1), *p* = 0.029), even though blood loss did not differ among the groups (1000 (500–1100) mL vs. 600 (350–1000) mL, *p* = 0.098). Postoperatively, patients with pre-existing liver pathologies stayed longer in the intensive care unit (3 (2–6) vs. 2 (1–3) days, *p* = 0.010), though interestingly this did not affect the overall duration of the hospital stay (Table 1).

As for postoperative complications, we did observe significantly more intraabdominal abscess formation in patients with advanced liver fibrosis and liver cirrhosis (40% vs. 11%, *p* = 0.006). In addition, they showed a higher mortality rate (20% vs. 3%, *p* = 0.021) than patients with no pre-existing liver pathologies. Both groups were comparable according to the Clavien–Dindo classification (Clavien major 60% vs. 51%, *p* = 0.372), and complications according to ISGPS such as PPH, PF and DGE did not differ between the two groups (Table 2).

In particular, specific parameters according to ISGPS, such as the first day of solid food intake, the duration of intraoperative administered nasogastric tube (NGT), the rate of reinsertion of an NGT, and the need for parenteral nutrition, also did not differ between the two groups (Table 3).

In the univariate analysis, the following factors qualified for multivariate analysis: preoperative diabetes mellitus, LC or advanced LF, reoperation, insufficiency of BDA, and suprafascial wound infection (Table 4). Note that DGE according to ISGPS was not statistically significant in the univariate analyses. Reoperation was a risk factor for high mortality in the multivariate analysis (*p* ≤ 0.001).

## 4. Discussion

To the best of our knowledge, this is the first study investigating common postoperative complications after PD in patients with LC and advanced LF according to the ISGPS definitions, with special emphasis on DGE, including DGE grading and specific parameters associated with DGE. There are several studies investigating postoperative complications and outcomes after PD in cirrhotic patients [16,17,33,34,35]. However, these studies show some limitations. First, and by far most importantly, almost all of the studies [16,17,33,34,35] include patient data before the year 2007—the year in which the International Study Group of Pancreatic Surgery developed a uniform definition of DGE, with DGE being further classified into three different grades (grade A, B and C) according to the clinical impact [29]. Thus, data were either only retrospectively classified according to ISGPS, or not at all, and sub-classification into grade A, B and C did not take place. One recent study by Cheng et al. investigated the outcome of minimally invasive PD; in their propensity matched analysis, no difference in DGE was found in patients with and without higher-grade liver pathologies [35]. The ISGPS definition of DGE and the sub-classification into these grades involves either the postoperative duration of an NGT or the need for the reinsertion of an NGT [29]. In our study, we could not detect a difference in respect to DGE classified by ISGPS, either in the general definition or according to the NGT-related parameters, such as the duration of intraoperative administered NGT or the rate of reinsertion of an NGT.

Reasons for impaired gastric motility in patients with LC may be autonomic dysfunction [22], higher postprandial glucose and insulin levels, and a lower secretion of ghrelin [23], a hormone that stimulates gastric emptying and is primarily produced in the stomach, but also to a lesser extent in other organs such as in pancreatic islet cells [36]. Hormone imbalance is also observed after PD, with decreased insulin secretion leading to hyperglycemia and increased GLP-1 secretion [37], both of which are known to delay gastric emptying [38]. Potential explanations for why we did not observe an even more severe DGE in patients with LC and advanced LF after PD have to remain speculative at this point, and might include imbalance of hormone regulation with decreased GLP-1 secretion in cirrhotic patients [39] as a counteraction to the increased GLP-1 secretion after PD per se. Furthermore, patients with LC and advanced LF might simply be well adapted to prolonged gastrointestinal transit, and thus, do not show more DGE after PD than patients without liver pathologies [40]. To clarify why we did not detect a difference between the two groups and to uncover a possible underlying mechanism, further studies are needed.

Patients with LC and advanced LF in our cohort showed a higher quantity of comorbidities measured by the CCI, and therefore were being put at higher perioperative risk, as represented by a higher ASA score. As a result, patients with LC and advanced LF to begin with were in worse condition than patients with healthy livers. Not surprisingly, postoperatively, they showed a higher mortality rate, which is in line with previous studies [16,17,18]. In order to minimize postoperative mortality in patients with LC and advanced LF as much as possible, patients should thus receive optimal supportive treatment to improve their medical status, such as parenteral nutrition, drainage of ascites, substitution of albumin, and prophylaxis of pneumonia by breathing exercises.

A correlation between erythrocyte concentrates that are transfused during surgery and mortality is well known [41,42], and according to current guidelines restrictive transfusion strategies are also recommended in patients with LC [43]. In our cohort, in patients with LC and advanced LF, more erythrocyte concentrates were transfused during surgery, despite the amount of blood loss not differing statistically between the two groups. One would assume a higher amount of blood loss in patients with LC and advanced LF; however, at our center, a standardized perioperative management—which involves preoperative consultation of our coagulation division and, if necessary, administration of vitamin K or fresh frozen plasma, intraoperative careful dissection, use of ligature, titan clips and ligature devices—might be a possible explanation for this. Eventually, patients with higher-grade liver pathologies are assumed to be of a higher bleeding risk, and therefore the threshold to transfuse is set lower by the attending surgeons and anesthesiologists.

Postoperatively, patients stayed at the intensive care unit (ICU) for at least one day; however, patients with LC and advanced LF spent significantly more days at the ICU. This is not surprising since they showed more comorbidities and a higher ASA score, are thereby considered to show higher morbidity and mortality after surgery per se, and simply might directly need prolonged postoperative intensive care [44]. Importantly, this did not affect the overall length of the hospital stay. Furthermore, patients with LC are known to show impaired immune function [45], which might also account for a longer stay at the intensive care unit. This impaired immune function might also explain the observation that patients with LC and advanced LF showed more intraabdominal abscess formation after surgery, even though rates of insufficiency of pancreatogastrostomy, biliodigestive anastomosis and duodenoenterostomy did not differ between the two groups. Moreover, there was no difference in intraoperatively taken microbiological samples of the gall fluid; however, patients with LC are known to show increased gut permeability and bacterial translocation [46], which not only accounts for the development of spontaneous bacterial peritonitis, but also might increase the risk of abscess formation after PD. Postoperative complications such as abscess formation—which was treated by either CT-guided percutaneous drainage or endoscopic ultrasound-guided transgastric drainage—did not lead to an increased need for a second surgery. This is in line with other studies that also did not report more reoperations after PD in cirrhotic patients compared to non-cirrhotic controls [16,17,18,19,34]. However, in our study, if reoperation occurred this was a risk factor for a higher mortality rate. Thus, patients with LC and advanced LF should be critically evaluated for PD. If in need of elective PD, risk factors known to influence surgical outcome in general for patients with compromised liver function should be carefully optimized (e.g., parenteral nutrition, drainage of ascites, substitution of albumin, prophylaxis of pneumonia by breathing exercises), and postoperative close surveillance to avoid reoperation should be mandatory.

## 5. Conclusions

In this study, we investigated the impact of common specific complications after PD in patients with pre-existing liver cirrhosis and advanced liver fibrosis. We showed that patients with pre-existing LC and advanced LF demonstrate higher morbidity and mortality, and thus, these patients should be critically evaluated for PD.

## Figures and Tables

**Table 1 jcm-10-02521-t001:** Demographic and perioperative data.

	Advanced Fibrosis and Cirrhosis	No Liver Pathology	*p*
	*n* = 15	*n* = 240	
age (a)	63 (57–77)	68 (59–75)	0.475
gender female	5 (3%)	106 (44%)	0.412
BMI	26.9 (23.7–28.1)	24.8 (22.7–28.1)	0.275
diagnosis malignant	9 (60%)	184 (77%)	0.210
weight loss	7 (47%)	129 (54%)	0.547
alcohol abuse	8 (53%)	62 (26%)	0.036
nicotine (active consumption)	6 (40%)	66 (28%)	0.378
preoperative biliary drainage	4 (27%)	114 (48%)	0.116
preoperative diabetes mellitus	5 (33%)	70 (29%)	0.773
Charlson Comorbidity Index	4 (3–5)	2 (2–3)	≤0.001
ASA physical status classification	3 (2–3)	2 (2–3)	0.016
duration of operation (min)	377 (330–471)	395 (315–467)	0.847
transfusions (erythrocyte concentrate)	2 (0–3)	0 (0–1)	0.029
blood loss (mL)	1000 (500–1100)	600 (300–1000)	0.098
positive intraoperative microbiology	5 (33%)	118 (49%)	0.071
venous resection	4 (27%)	45 (19%)	0.499
multivisceral resection	1 (7%)	13 (5%)	0.590
single loop reconstruction	13 (87%)	202 (84%)	1.000
infracolic reconstruction	3 (20%)	77 (32%)	0.381
retrocolic duodenoenterostomy	12 (80%)	199 (83%)	1.000
pylorus-preserving procedure	11 (73%)	202 (84%)	0.282
stay in hospital (d)	25 (20–36)	23 (17–30)	0.392
stay in intensive care unit (d)	3 (2–6)	2 (1–3)	0.010
stay in intensive care unit with respirator (d)	0 (0–1)	0 (0)	0.216

Data are shown as frequency (%) or median (interquartile range), BMI = body mass index, ASA = American Society of Anesthesiologists.

**Table 2 jcm-10-02521-t002:** Postoperative outcome/complications.

	Advanced Fibrosis and Cirrhosis	No Liver Pathology	*p*
	*n* = 15	*n* = 240	
PPH grade B/C	5 (33%)	61 (25%)	0.545
PF grade B/C	3 (20%)	45 (19%)	0.733
insufficiency of BDA	2 (13%)	13 (5%)	0.196
insufficiency of DE	0 (0%)	8 (3%)	1.000
wound infection (suprafascial)	3 (20%)	45 (19%)	1.000
intraabdominal abscess formation	6 (40%)	26 (11%)	0.006
reoperation	2 (13%)	30 (13%)	0.693
Clavien major (grade III–IV)	9 (60%)	115 (48%)	0.372
mortality	3 (20%)	8 (3%)	0.021
delayed gastric emptying	6 (40%)	121 (50%)	0.932
grade A	3 (20%)	67 (28%)	1.000
grade B	3 (20%)	34 (14%)	0.396
grade C	0 (0%)	21 (9%)	0.607

Data are shown as frequency (%), PPH = postpancreatectomy hemorrhage, PF = pancreatic fistula, BDA = biliodigestive anastomosis, DE = duodenoenterostomy.

**Table 3 jcm-10-02521-t003:** Delayed gastric emptying (DGE) parameters according to ISGPS.

	Advanced Fibrosis and Cirrhosis	No Liver Pathology	*p*
	*n* = 15	*n* = 240	
first day of solid food intake	10 (7–14)	9 (7–15)	0.901
intraoperative gastric tube (d)	4 (3–8)	4 (3–7)	0.812
reinsertion of gastric tube	1 (7%)	65 (27%)	0.072
parenteral nutrition (d)	3 (0–9)	3 (0–7)	0.643

Data are shown as frequency (%) or median (interquartile range).

**Table 4 jcm-10-02521-t004:** Risk factors associated with high mortality.

	Odds Ratio	95%-CI	*p*
**univariate**			
preoperative diabetes mellitus	3.043	0.899–10.299	0.087
advanced fibrosis and cirrhosis	7.250	1.704–30.853	0.021
reoperation	36.500	7.327–181.833	≤0.001
insufficiency of BDA	8.286	1.902–36.098	0.016
wound infection (suprafascial)	3.876	1.131–13.283	0.037
DGE grade B/C	2.067	0.478–8.926	0.390
**multivariate**			
reoperation	26.899	4.067–177.915	≤0.001

CI = confidence interval.

## Data Availability

Our anonymized pancreatic resection database contains sensitive data (e.g., date of surgery), with which certain patients could be identified. According to German law and according to the approval of the ethics committee, these data must not be published. Access to the database can be obtained from the corresponding author upon reasonable request.

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
