# Peer review of "Chronic Liver Disease Increases Mortality Following Pancreatoduodenectomy"

_jcm, 2021, doi:10.3390/jcm10112521_

Round 1

Reviewer 1 Report

The manuscript is well written and well structured.

It is an original article about the influence of pre-existing liver disease on delayed gastric emptying (DGE) after pancreatoduodenectomy (PD).

It shows no statistically significant difference in DGE after PD between patients with advanced fibrosis or cirrhosis and patients without liver disease.

The impact of other common specific complications after PD according to the ISGPS definitions were also described and evaluated.

Data from the literature are restricted and further studies are needed.

The topic is interesting and in line with the journal.

Reviewer 2 Report

The authors present a well written manuscript retrospectively comparing the short term outcomes of pancreatoduodenectomy in liver-healthy and cirrhotic/fibrotic patients. The authors report several interesting differences between the two groups such as a significantly higher mortality (205 vs. 3%), more intraabdominal abscess formation with no difference in PPH or PF.

Interestingly, no significant difference was found between the two groups in regard to DGE, which is also stated in the title of the manuscript. This however, is the biggest problem of the manuscript: If a test result is not significant, either there is actually no effect or an existing effect could not be detected. Therefore, it must not be concluded from non-significant test results that there is no difference! Applied to the manuscript that would mean that a difference in DGE between liver healthy/cirrhotic patients could not be detected. It might be there (a would only be detected in e.g. a larger sample size) or there really is no difference. (thus p5L180f cannot be stated as it is)

I would like to recommend to the authors to change the manuscript accordingly and to put more emphasis (including the title) on all the other differences between the two groups. In addition, it might be useful to do a multivariate analysis with DGE as a target and see what influencing factors (PF, abscess formation etc.) remain in the multivariate model and if liver texture is one of the multivariately independent factors for DGE. 

Minor points:

  • p1L29: In contrast to septic complications, most probably, a longer hospital stay due to DGE will not influence long-term cancer-specific survival.
  •  p3L105: The multivariate model should be better characterised:  p-value criteria for inclusion or removal from the multivariate model in each step should be reported, as well as forward/backward, likelihood ratio
  • The evaluation of liver fibrosis/cirrhosis should be presented better than in p2L68/69. Especially as there is a rather grey are between NASH and (advanced) fibrosis from a purely clinical point of view
  • p6L233f: "this was considered a risk factor": you mean "this was shown to be a risk factor (in our data)"?
  • p7L240: "still be critically evaluate for PD": evaluated
  • p7L241: I agree that PD, especially in high risk patients should be performed in high volume centers. However, this conclusion cannot be drawn from your data, can it?

Round 2

Reviewer 2 Report

The authors have adequately addressed all the issues raise by the reviewers- I have no more concerns and would like to congratulate them on the excellent paper.